# Faecal Microbiota Composition Varies between Patients with Breast Cancer and Healthy Women: A Comparative Case-Control Study

**DOI:** 10.3390/nu13082705

**Published:** 2021-08-05

**Authors:** Christine Bobin-Dubigeon, Huyen Trang Luu, Sébastien Leuillet, Sidonie N. Lavergne, Thomas Carton, Françoise Le Vacon, Catherine Michel, Hassane Nazih, Jean-Marie Bard

**Affiliations:** 1Institut de Cancérologie de l’Ouest, 44805 Saint-Herblain, France; jean-marie.bard@ico.unicancer.fr; 2EA 2160—IUML FR3473 CNRS, Nantes University, 44035 Nantes, France; tranglh269@gmail.com (H.T.L.); el-hassane.nazih@univ-nantes.fr (H.N.); 3Research Center of Human Nutrition CRNH Ouest, 44093 Nantes, France; catherine.michel@univ-nantes.fr; 4Biofortis Mérieux NutriSciences, 3 Route de la Chatterie, 44800 Saint-Herblain, France; sebastien.leuillet@mxns.com (S.L.); sidolavergne@hotmail.com (S.N.L.); thomas.carton@mxns.com (T.C.); francoise.le.vacon@mxns.com (F.L.V.); 5UMR 1280, 44035 Nantes, France

**Keywords:** microbiota, breast cancer, healthy women, 16S rRNA, metabarcoding analyses, qPCR, abundance

## Abstract

The intestinal microbiota plays an essential role in many diseases, such as obesity, irritable bowel disease (IBD), and cancer. This study aimed to characterize the faecal microbiota from early-stage breast cancer (BC) patients and healthy controls. Faeces from newly diagnosed breast cancer patients, mainly for an invasive carcinoma of no specific type (HR^+^ and HER2^−^), before any therapeutic treatment and healthy controls were collected for metabarcoding analyses. We show that the Shannon index, used as an index of diversity, was statistically lower in the BC group compared to that of controls. This work highlights a reduction of microbial diversity, a relative enrichment in Firmicutes, as well as a depletion in Bacteroidetes in patients diagnosed with early BC compared to those of healthy women. A tendency towards a decreased relative abundance of *Odoribacter* sp., *Butyricimonas* sp., and *Coprococcus* sp. was observed. This preliminary study suggests that breast cancer patients may differ from healthy subjects in their intestinal bacterial composition.

## 1. Introduction

Breast cancer (BC) is the most commonly diagnosed solid organ cancer in women worldwide. In 2015, it represented 29% of all new cancers among women, with more than 500,000 deaths annually overall [1]. Apart from the genetic predispositions that cause 5 to 10% of BC, other risk factors for its development have been recognized, such as obesity in postmenopausal women, high oestrogen exposure, and diet [2]. The recent investigations of the influence of the gut microbiota on human health, in particular on metabolism and nutrient absorption in the digestive tract, as well as on many diseases (e.g., obesity, diabetes, and allergies), raise the question of whether this community is involved in the aetiology of cancer [3], especially in BC. Answering this question could lead to new preventative strategies for this pathology.

Our gut microbiota with its one hundred trillion bacteria is now considered as an additional organ of the human body. This intestinal flora is able to provide a protective effect against invading pathogens to help with the development of the immune system and with the recovery and absorption of nutrients and non-nutritive compounds. However, the disruption of the subtle equilibrium of this commensal community can lead to microbial dysbiosis and the rise of potentially pathogenic bacteria, some of which could be carcinogenic. Evidence for the cancer-promoting effects of intestinal dysbiosis has been well documented in colorectal, liver, and pancreatic cancer [4,5,6,7]. In BC, it has been reported that experimental alterations of the gut microbiota could influence the generation of hyperplastic and neoplastic lesions in the mammary glands [8]. These results are compatible with the findings of epidemiological studies that suggest an association between microbial dysbiosis due to antibiotic therapy and an increased incidence in breast carcinoma [3,9].

In a previous study, we highlighted the modifications in the gut microbiota composition between BC patients at different stages [10]. Therefore, we hypothesized that the gut microbiota differed between patients with BC and healthy women. To test this hypothesis, we characterized and compared the microbiota in stool samples from patients with early BC prior to starting their therapy and faeces from control women, using 16S rRNA gene sequencing. In a second step, specific bacterial genera were quantified in the two cohorts, by using quantitative RT-PCR technique.

## 2. Materials and Methods

### 2.1. Recruitment and Sample Collection

This study included 30 healthy women controls (no pathology declared, no current medication except contraceptives or menopausal hormone therapy) and 25 BC patients before any anticancer therapy was started. Patients had been referred to our hospital (ICO René Gauducheau, Saint-Herblain, France) for early-stage BC. All medications well known to impact microbiota were considered to be exclusion criteria (such as antibiotics or gastrointestinal drugs). For all participants, we recorded age; weight; size; menopausal status; lifestyle, such as smoking and alcohol consumption; oral contraceptives; and menopausal hormone therapy. For patients, we also included the type of BC, the histo-prognostic grade, the clinical stage, and the tumour localization and size. Informed consent was obtained from patients and control women to use their biological specimens and clinicopathological data for research purposes, as required by the French legislation (CRB-Tumorothèque–ICO, France) [11]. Figure 1 describes the design of the study. 

The patients and healthy women directly defecated into a sterile box at home. Whole stools collected were immediately stored at 4 °C; 2 g of fresh faeces were processed for qPCR, and frozen at the hospital at −80 °C for metabarcoding analysis.

### 2.2. V3–V4 16S rRNA Gene Sequencing

Genomic DNA was isolated using a Maxwell^®^ 16 Tissue DNA Purification Kit (Promega Corporation, Madison, WI, USA). DNA extraction was performed from one aliquot of 200 mg of frozen faecal sample (30 samples from healthy women and 25 from BC patients). Total genomic DNA was collected in a volume of 200 µL. Double-stranded DNA (dsDNA) concentrations were measured by fluorimetry using the Qubit^®^ 2.0 Fluorometer and the Qubit^®^ dsDNA broad range assay (Invitrogen by Life Technologies, Carlsbad, CA, USA).

PCR amplification was performed using 16S universal primers 341F and 785R targeting the V3–V4 region of the bacterial 16S ribosomal genes [12]. The 16S V3–V4 amplicon size was verified by capillary electrophoresis (Agilent 2100 Electrophoresis Bioanalyzer Instrument, Agilent technologies, Santa Clara, CA, USA). All amplicons were purified with magnetic beads (Agencourt AMPure XP beads, Beckman coulter, Brea, CA, USA). Then, for each sample, a sequencing library was generated by addition of dual indices and Illumina sequencing adapters, using a Nextera XT Index kit (Illumina, San Diego, CA, USA). Each library was cleaned with magnetic beads and its size was determined by capillary electrophoresis. After quantification by fluorimetry (Qubit^®^ 2.0 Fluorometer), libraries were normalized and pooled. The pool of libraries was further denatured and sequenced on the Illumina MiSeq platform, using a 2 × 250 paired-end Miseq kit V2 (Illumina, San Diego, CA, USA).

### 2.3. Bioinformatic Analysis

Read sequences from faecal microbiota were analysed using an in-house bioinformatic pipeline based on mothur v1.33.3 software [13]. Briefly, sequences were trimmed and aligned to the V3–V4 region of the 16S gene of the Greengenes database that had been formatted with mothur (gg_13_5_99 release). Chimera sequences were removed using the UCHIME algorithm. Reads were classified using a naive Bayesian classifier against RDP database release 14 with a bootstrap cut-off of 60%. Sequences were then clustered into operational taxonomic units (OTUs) using furthest-neighbour clustering at a similarity threshold of 97%. For each sample, the OTU-based microbial diversity was estimated by calculating the Shannon index, and rarefied Chao1 indices (to 10,000 reads) were then calculated.

### 2.4. Quantitative PCR (qPCR) for Bacterial Copy Numbers

All aliquots of fresh faeces (2 g) from 25 BC patients and control women (*n* = 29, one sample being too small to allow for this analysis) were prepared and used for the bacterial DNA extraction with protocols previously described [10], and immediately frozen at −80 °C. Real-time PCR was then performed to enumerate the copy numbers of total bacteria and some selected bacterial populations (Bacteroidetes phylum, Firmicutes phylum, *Lactobacillales* sp., *Clostridium* cluster IV, *Faecalibacterium prausnitzii*, *Clostridium* cluster XIVa, *R. intestinalis*, *Blautia* sp., *Lactonifactor longoviformis*, *Bifidobacterium* sp., Coriobacteriaceae, *E. lenta*, *Escherichia* and *Shigella*), using the specific primers also presented in our previous publication. The number of bacteria for each bacterial population was expressed as log10 equivalent bacteria (noted log10 eq. bact.) per gram of fresh stool. 

### 2.5. Statistical Analysis

For V3–V4 16S rRNA gene sequencing, richness and diversity indices and relative abundances of taxa at the phylum, family, and genus taxonomic levels, were compared between patients and healthy women using an ANOVA model, including adjustment on age (analysis of covariance). A Benjamini Hochberg procedure was applied to control the false discovery rate (FDR) due to multiple hypotheses tests on all taxa at each taxonomical level. The *p*-values are given with FDR and without FDR (*FDRpadj and FDRpnonadj*, respectively). No adjustment was made for the analysis of richness and diversity. For all statistical tests (two-tailed), the 0.05 level of significance was used to justify a claim of a statistically significant effect. Statistical analyses were performed by BIOFORTIS using the SAS^®^ software version 9.3 (SAS Institute Inc., Cary, NC, USA). Graphical representations of the microbiota data were generated using the R software version 3.2.3 (R Core Team, 2016).

For qPCR analysis, as described in Luu et al. [10], all statistics were run using the SAS software version 9.3 (Chapel Hill, NC, USA). Bacterial gene counts were computed and compared between groups of patients using the Wilcoxon test.

The comparison of anthropometric data, lifestyle parameters, and medication consumption between groups were performed using the Wilcoxon test or Fisher test according to the nature of the variable.

## 3. Results

### 3.1. Clinical Characteristics of the Studied Population

The main characteristics of the two studied cohorts are described in Table 1.

Patients and healthy women controls were comparable for menopausal status and BMI. However, patients were significantly older than healthy controls (median (25th–75th): 63.0 (53.8–71.0) vs. 53.5 (47.0–62.0) years old; *p* = 0.01). Lifestyle appeared similar in the two groups according to smoking and alcohol consumption. The use of menopausal treatment or hormonal contraception was also the same in patients and controls.

Breast cancer patients were mainly diagnosed for an invasive carcinoma of no specific type (76%), clinical stage I (84%), with a good prognostic grade (20% of grades I and 64% grade II), 100% being ER/Pg+ and HER2- (Table 2). The tumour localization was mainly unifocal, with a median size of 12.5 mm. No carriers of the tumour *BRCA1* and/or *BRCA2* mutations were identified in the patient group, as described before [10].

### 3.2. Faecal Microbiota Analysis

#### 3.2.1. qPCR for Specific Bacterial Copy Numbers

In a previous work, we highlighted that faecal microbiota composition of breast cancer patients differs according to clinical characteristics by using qPCR 16S rRNA sequences targeting bacterial groups of interest. This approach was therefore applied to the healthy control group (only *n* = 29 available). The 16S rRNA gene-targeted specific primers per bacterial group/species are described in [10], and the relative abundances of bacteria are summarized in Table 3. No difference was observed between the two groups according to age with the Wilcoxon test. 

No significant differences were found in the number of total bacteria observed between the BC patient group and healthy control group (data not shown). The relative abundance expressed as median (25th–75th) of the Bacteroidetes and Firmicutes phyla significantly differed between the two groups. We found that the relative abundance of Firmicutes was significantly higher in patients with 36.10% (24.91–43.46) when compared to that of healthy controls 15.46% (10.75–33.57) (*p* = 0.003). In contrast, the Bacteroidetes phylum was significantly more relatively abundant in controls than in it was in patients, accounting for 19.14% (16.27–22.75) vs. 14.24% (12.52–19.21), respectively (*p* < 0.001). At the family level, no significant differences were found in Coriobacteriaceae abundance observed between the BC patient group and healthy control group.

*Clostridium* cluster IV and cluster XIVa were in a higher relative abundance in the BC group (12.02% (9.33–21.56) vs. 6.1% (3.63–11.91) *p* = 0.003) and 18.64% (10.94–22.43) vs. 5.97% (4.02–15.59); *p* = 0.012), respectively. At the genera level, the relative abundance of *Blautia* was significantly higher in the BC group compared to that of controls.

#### 3.2.2. V3–V4 16S r RNA Gene Sequencing

α-diversity analysis

The α-diversity of faecal microbiota in healthy and BC groups identified by 16S rRNA gene sequencing is described in the Figure 2. The β-diversity was analysed by principal coordinates analysis (data not shown), the different samples were not partitioned according to groups at the phylum, family and genera levels.

While the rarefied Chao1 index, a marker of richness, was not statistically different between the two groups (Figure 2a—2674 ± 637 vs. 2382 ± 540, *padj* = 0.16, *pnonadj* = 0.08), the Shannon index, a marker of richness and evenness, of BC patients is lower than in healthy control (Figure 2b—5.18 ± 0.42 vs. 4.89 ± 0.40, *padj* = 0.04, *pnon adj* = 0.008), suggesting a significantly lower diversity in our BC patients compared to our healthy individuals (Figure 1).

Taxonomic analysis of faecal microbiota composition at the phylum, family, and genus levels

The barplots of relative abundances at phylum levels for individual data (a) or means per condition (b) are presented in Figure 3. The relative abundances at the phylum level expressed as mean (SEM) in the control group vs. BC group were 3.40% (0.51) vs. 3.62% (0.56); 2.89% (0.37) vs. 2.13% (0.41); 2.23% (1.05) vs. 5.15% (1.15) for Actinobacteria, Proteobacteria, and Verrucomicrobia, respectively. According to this taxonomic analysis at the phylum level, we found that Firmicutes were the most predominant phylum in both groups, with a significantly higher relative abundance in patients 61.64% (1.99) compared to controls 54.19% (1.81) (*FDRpadj* = 0.03, *FDRpnonadj* = 0.01). In contrast, the Bacteroidetes phylum was significantly more abundant in healthy controls than in BC patients, accounting for 36.98% (1.84) and 27.19% (2.02), respectively (*FDRpadj* = 0.03, *FDRpnonadj* = 0.005) (Figure 3b). 

The barplots of relative abundances at the family level for individual data (a) or means per condition (b) are presented in Figure 4. At the family level, no significant differences were observed between the two groups.

The heatmap of relative abundances at the genus level is presented Figure 5. Among bacterial genera, *Coprococcus* (Firmicutes), *Odoribacter*, and *Butyricimonas* (Bacteroidetes) were found in less relative abundance in breast cancer patients compared to that of controls (1.36% (0.29) vs. 2.84% (0.27) *FDRpadj* = 0.11, *FDRpnonadj* = 0.007; 0.14% (0.04) vs. 0.30% (0.04), *FDRpadj* = 0.07, *FDRpnonadj* 0.003and 0.1% (0.05) vs. 0.32% (0.04), *FDRpadj* = 0.07, *FDRpnonadj* 0.003, respectively). Conversely, *Clostridium* XVIII and *Lachnospira* (Firmicutes) were in a higher relative abundance in BC patients (0.77% (0.11) vs. 0.3% (0.10), *FDRpadj* = 0.11, *FDRpnonadj* 0.008; and 1.23% (0.17) vs. 0.58% (0.16), *FDRpadj*= 0.08, *FDRpnonadj* 0.003).

## 4. Discussion

The human microbiota is involved in the development and progression of major human disorders. Changes in composition and function of the microbiota have been highlighted in many human disorders and especially in patients with colorectal cancer, liver cancer, or prostate cancer [14,15]. According to epidemiologic data, the human microflora could contribute to around 15% of cancer cases worldwide [1]. Furthermore, intestinal microbiota dysbiosis has been described in many human digestive diseases such as IBD, metabolic diseases, and extra digestive diseases such as neuro-psychiatric diseases (Parkinson and autism) or allergies [7,16].

Recent data suggest a relationship between breast cancer and microbiota, especially with intestinal microbiota, although the role of mammary microbiota in modulating breast cancer development remains to be clarified [17]. This link has been explored in our previous work [10], where we highlighted that intestinal microbiota composition in BC patients differed according to clinical characteristics and BMI. More recently, Goedert et al. [18] have shown that faecal microbiota profiles differ between BC and healthy individuals, showing that postmenopausal women with breast cancer have an altered composition and oestrogen-independent low diversity of their gut microbiota. However, the association between gut microbiota dysbiosis and the risk of breast cancer should be in part clarified by clinical studies, such as those in progress in Italy [17]. To better understand the relationship between BC and microbiota, the faecal microbiota of our previously described BC cohort [10] was compared to a control group of healthy women. As described in our previous paper, clinical and biological characteristics of early breast cancer ER/Pg+ were mostly common in cohorts of French women, in term of carcinologic characteristics, age, and BMI. Menopausal status was the same in the two groups. However, the low number of patients and controls does not allow us to draw definitive conclusions on this point, regarding the not significant, but relatively low *p* value (0.10). This is of particular importance, since microbiota plays a role in oestrogen metabolism, while cancer risk in postmenopausal women is directly related to the level of endogenous oestrogens [19]. 

The two studied cohorts had the same smoking status and alcohol consumption. Alcohol consumption is positively associated with an increase of BC risk, through oestrogen metabolism but also through modification of microbiota composition in animals and in humans [20].

Our control group was significantly younger than was our patient group, which may have interfered with our results, since age has been shown to be one of the factors causing changes in the intestinal microbiota composition [21,22]. However, while this was statistically significant, the difference between the two groups was around 10 years. It should be kept in mind that changes in gut microbiota were mainly observed in elderly populations (>65 years [21,23]) In our study, only 28% of BC patients were older than 65.

Age and ageing can impact the composition of the intestinal microbiota, but according to [24], these changes are mainly due to health status, medication drugs, and also lifestyle factors. In a cross-sectional study in Europe, the authors of [25] concluded that the variation of abundance of the different phylogenetics groups in faecal microbiota was mainly due to dietary habits rather than age.

However, to be sure that this parameter did not interfere with the comparison of microbiota in our case/control study, the ANOVA model was built to include adjustment on age, as described previously.

It is a common knowledge that species diversity of a bacterial community is characterized by both species’ richness and evenness [26]. In the present study, we observed a lower but nonsignificant richness of gut microbiota in patients versus that of controls using the Chao1 index. However, we observed a significant loss of diversity in BC patients compared to that of healthy controls when using the Shannon index; this index includes both richness and evenness. Our results are in agreement with previously published case control data with postmenopausal women that showed the same lower diversity in a cohort with the same characteristics, even if the authors found a significant difference for Chao1 and a nonsignificant difference for the Shannon index [18]. A reduction in intestinal bacterial diversity has already been reported in other cancers (e.g., colorectal) and other pathologies (e.g., inflammatory bowel disease, obesity, and autism) [3,18,27,28,29]. 

Firmicutes and Bacteroidetes are the two main phyla involved in the colonic metabolism of indigestible nutrients, including dietary fibres and polyphenols. Similar to results previously described in different pathologic situations [30,31], the relative abundance of Bacteroidetes in healthy faecal microbiota were lower than that of Firmicutes, both with NGS and qPCR tools, despite large interpersonal variations. Moreover, the relative abundance of these two main phyla also changes with age, even in healthy subjects, as shown in [32]. The mechanisms underlying these changes in relative abundance, in relation to breast cancer pathology and age, need to be clarified by larger-scale studies.

At the genera level, we demonstrated in our previous study that a high abundance and relative proportion of some bacterial groups such as *Clostridium* cluster XIVa, *Clostridium* cluster IV, *Faecalibacterium prausnitzii*, and *Blautia* sp. was associated with a more severe clinical stage in BC patients [10]. In agreement with these findings, the bacterial groups *Clostridium* cluster XIVa and *Clostridium* cluster IV also seemed to be enriched in the gut microbiota of BC patients in the present study; however, the difference was not statistically significant when adjusted for age. As mentioned earlier, the potential of gut microbiota to impact the oestrobolome, defined as “the aggregate of enteric bacterial genes whose products are capable of metabolizing estrogen has been hypothesized as a possible link between this community and the development of breast carcinogenesis [33]. It has been reported that gut microbial deconjugating enzymes (β-glucuronidases and β-glucosidases) can catalyse the hydrolysis of inactive glucuronidated oestrogens, thus promoting the reabsorption of their free active forms into the enterohepatic process [34]. The potential of intestinal bacteria that possess deconjugating enzymatic activity (β-glucuronidase and β-glucosidase) to modulate oestrogen reabsorption has been suggested as a possible hypothesis connecting the metabolic effects of gut microbiota with the development of BC [35]. These enzymatic activities might be harmful due to the established correlation between serum levels of oestrogen and BC risk. Most β-glucuronidase- and β-glucosidase-positive bacteria belong to two clusters: *Clostridium* cluster XIVa and *Clostridium* cluster IV, such as *Clostridium*, *Faecalibacterium*, *Ruminococcus*, and *Roseburia* genera but also *E. coli* [34]. Thus, the gut microbiota enriched with *Clostridium* cluster XIVa and *Clostridium* cluster IV could represent a significant risk factor in women. However, Goedert JJ et al. [18] showed that altered composition and lower diversity of the microbiota of BC postmenopausal women are independent from oestrogen.

Another observation concerned the nonsignificant decrease of the relative abundance of *Bifidobacterium* in faecal microbiota of BC patients. *Bifidobacterium* genera has long been used in probiotics that exert numerous beneficial effects in overweighted BC survivors [17], such as improvement of symptoms of eczema and irritable bowel syndrome [36]. Interestingly, Sivan et al. [37] showed that the oral administration of *Bifidobacterium* alone can improve the immune response against melanoma growth in several different mouse models. More recently, an interventional clinical study in BC patients suggested a positive influence of *Bifidobacterium* supplementation on the diversity of faecal microbiota [17]. 

Our results suggested a trend to a decreased abundance of *Coprococcus*, *Butyricimonas,* and *Odoribacter* genera in our BC patients. A similar result was described for *Coprococcus* in colorectal cancer patients [38], but also in autistic children [26]. In addition, the abundance of *Coprococcus* was reduced in mice exposed to social disruption stress, which correlated to circulating levels of stressor-induced pro-inflammatory cytokines [39]. *Butyricimonas* has shown a similar lower level in multiple sclerosis patients vs. healthy volunteers [40] and has been described to be decreased in non-Hodgkin lymphoma at high risk of bloodstream infection compared to that of patients with low risk of infection [41]. Lower levels of *Odoribacter* have been observed in patients suffering from rectal carcinoma [42]. *Coprococcus* (propionate and butyrate), *Odoribacter* (butyrate), and *Butyricimonas* (butyrate) are all short chain fatty acids (SCFA)-producer genera. SCFA have been shown to interfere in multiple ways in breast cancer, such as cell proliferation, apoptosis, or gene expression [43]. 

It is too early to hypothesize that the suggested decrease in *Bifidobacterium*, *Coprococcus*, *Butyricimonas,* and *Odoribacter* genera observed in our BC patients at an early stage of their disease might be linked to a role of these bacteria in the development of their cancer.

The role of human gut microbiota in BC could affect breast cancer risk through several ways by its ability to modulate oestrogen metabolism, inflammation, and immunity. Indeed, the gut microbiota can induce breast tumorigenesis as well as modulate the inflammatory response. Thus, it has been demonstrated that lipopolysaccharide, a component of the gram-negative bacterial cell wall, can trigger a systemic inflammatory process with production of pro-inflammatory cytokines, such as interleukin (IL)-6, IL-1, tumour necrosis factor alpha, and interferon gamma [44]. A higher level of inflammation linked to these changes in bacteria could potentially contribute to the development of breast cancer [45]. This hypothesis fits with the reduced incidence of breast cancer with nonsteroidal anti-inflammatory drug consumption [24,26,27]. 

The composition of the faecal microbiota is influenced by multiple factors, such as lifestyle, dietary factors, and co-morbidities. The role of each of these parameters remains unclear and would require studies that allow for multivariate analysis. 

In breast cancer, the faecal microbiome differs according to the tumour grade (previous published work [10]) and the receptor status [46]. Therefore, larger clinical studies should be considered to clarify the role of these carcinological characteristics on the composition of the microbiota in patients. In addition, the exploration of the link between faecal microbiota and breast cancer should be extended to the study of all microbiomes, in particular the tumoral microbiota, which is especially rich and diverse [47] and dependant on cancer subtype [20].

## 5. Conclusions

The present study highlighted a reduction of microbial diversity; a relative enrichment in Firmicutes, *Clostridium* cluster XIVa, and *Clostridium* cluster IV; and a depletion in Bacteroidetes, *Bifidobacterium* sp., *Odoribacter* sp., *Butyricimonas*.sp., and *Coprococcus* sp. in patients diagnosed with early BC compared to those of healthy women. 

This preliminary study suggests that breast cancer patients differ from healthy subjects in their intestinal microbial composition. Further studies are needed to explore the relationship between microbiome and breast cancer and understand the mechanisms involved.

## Figures and Tables

**Figure 1 nutrients-13-02705-f001:**
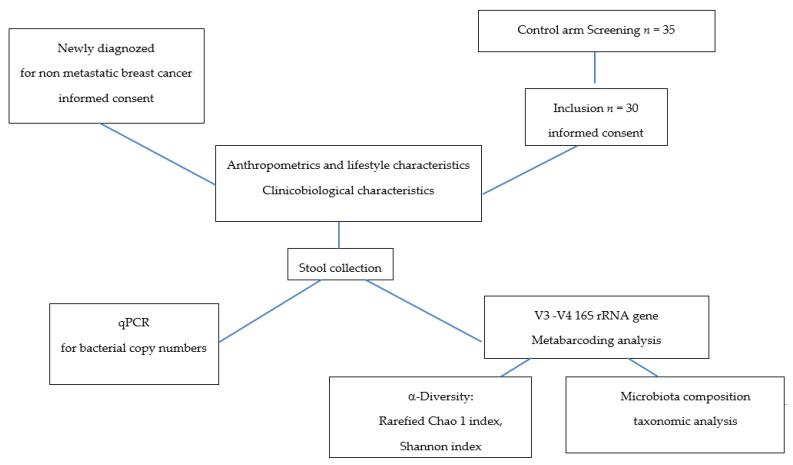
Study design.

**Figure 2 nutrients-13-02705-f002:**
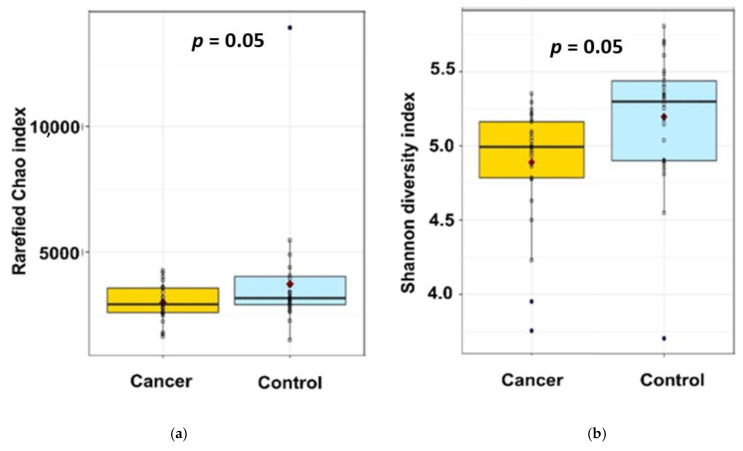
The α-diversity of faecal microbiota in the 2 studied groups identified by 16S rRNA gene sequencing: (**a**) Rarefied Chao1 index; (**b**) Shannon index. Indicated *p* values are not adjusted on age.

**Figure 3 nutrients-13-02705-f003:**
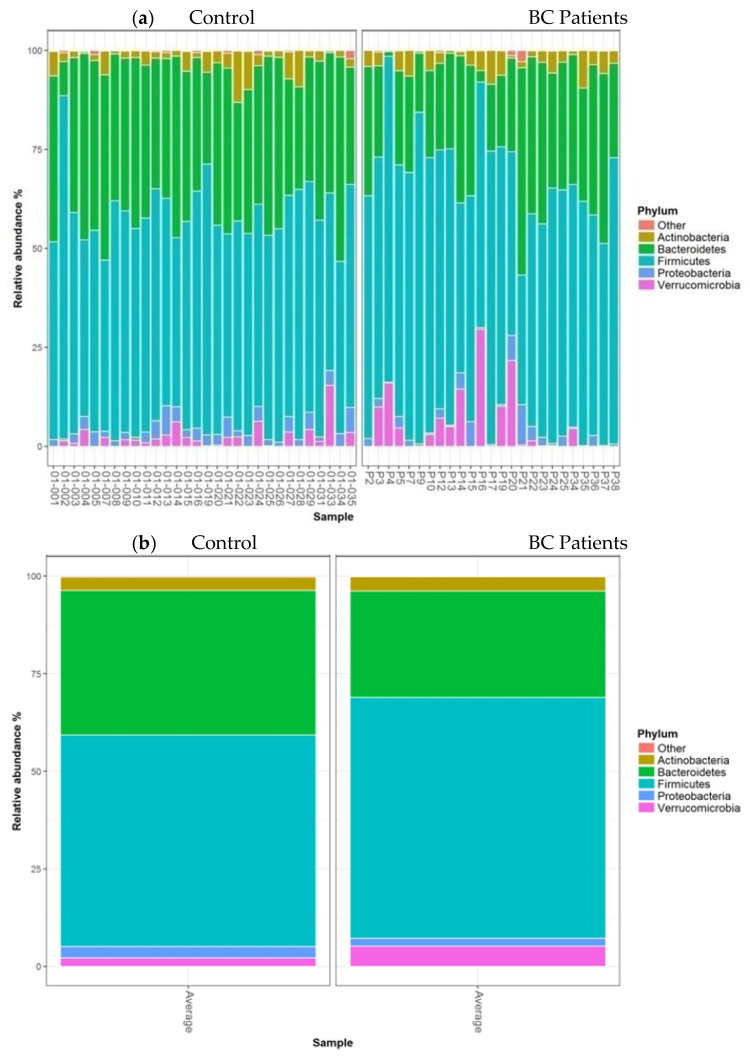
Barplots of relative abundances at the phylum level for individual data (**a**) or means per condition (**b**). Only taxa present on average in all samples at a threshold ≥0.5% or present in at least 10% of samples at a threshold ≥0.5% are individually represented.

**Figure 4 nutrients-13-02705-f004:**
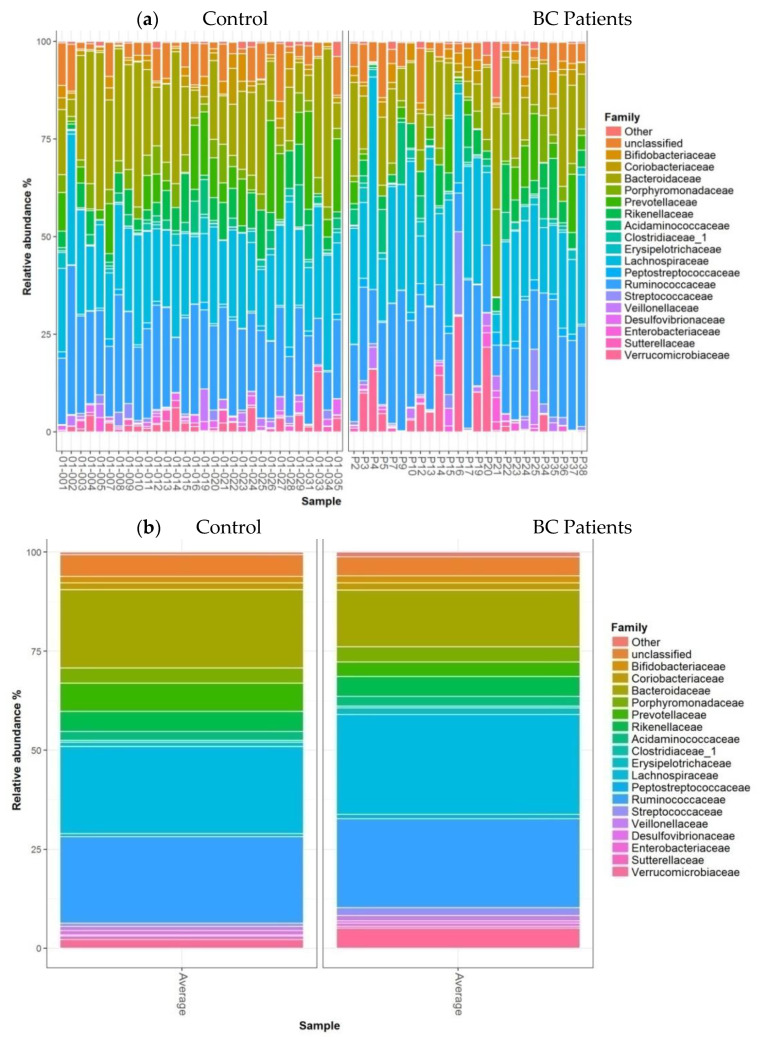
Barplots of relative abundances at the family level for individual data (**a**) or means per condition (**b**). Only taxa present on average in all samples at a threshold ≥0.5% or present in at least 10% of samples at a threshold ≥0.5% are individually represented.

**Figure 5 nutrients-13-02705-f005:**
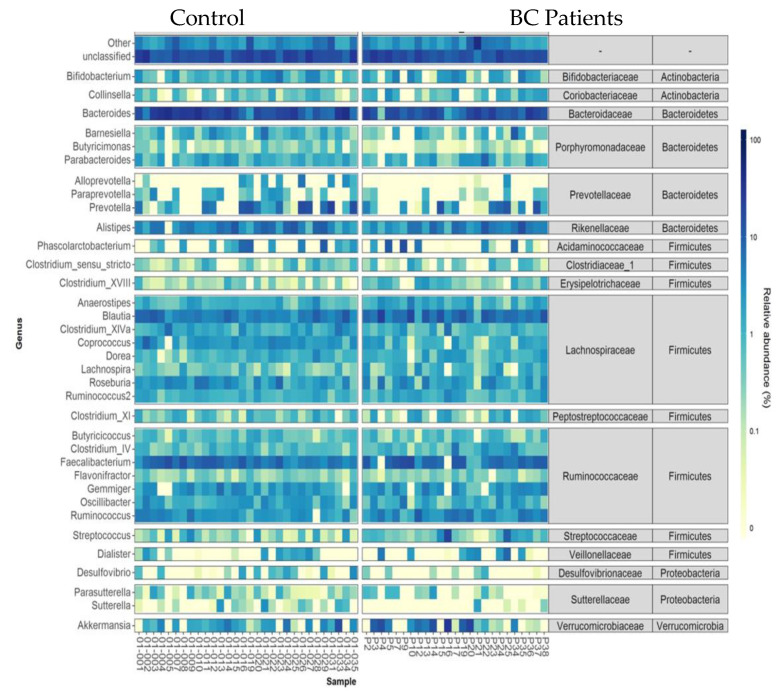
Heatmap of relative abundances at the genus level for individual data. Only taxa present on average in all samples at a threshold ≥0.5% or present in at least 10% of samples at a threshold ≥0.5% are individually represented. Taxa present at a threshold ≤0.01% in a sample were artificially considered as absent in this sample for this graphical representation.

**Table 1 nutrients-13-02705-t001:** Description of studied populations.

Characteristics	Healthy Controls (*n* = 30)	BC Patients (*n* = 25)	*p*
Age (year) *	53.5 (7.0–62.0)	63.0 (53.8–71.0)	0.01
BMI (kg·m^−2^) *	23.8 (22.4–25.6)	24.0 (21.5–26.4)	0.85
Menopausal status **	20 (66.6)	22 (88.0)	0.10
Alcohol consumption			1
<7 glasses/w	29 (96.6)	24 (96.0)	
>10 to 14/w	1 (3.3)	1 (4.0)	
smoking			1
0	28 (93.3)	25 (100)	
1–5/d	1 (3.3)	0	
6–10/d	1 (3.3)	0	
>20/d	0	0	
Medication treatment			
Hormone therapy	4 (13.3)	7 (28.0)	0.20

* Median (25th–75th) ** Frequency (percentage).

**Table 2 nutrients-13-02705-t002:** Carcinologic characteristics of the BC cohort (*n* = 25).

Type of Cancer	
Invasive Carcinoma of No Special Type (Ductal)	19 (76.0%)
Invasive Lobular Carcinoma	6 (24.0%)
Histoprognostic Grade	
Grade I	5 (20.0%)
Grade II	16 (64.0%)
Grade III	4 (16.0%)
Clinical Stage	
I	21 (84.0%)
IIa	3 (12.0%)
IIIb	1 (4.0%)
Tumoral Overexpression Receptors	
ER/Pg+	25 (100%)
HER2+	0
Tumour Localization	
unifocal	23 (92.0%)
multifocal	2 (8.0%)
size (mm) *	12.5 (9.0–14.7)
Gene *BRCA1/BRCA2*	0

Expressed as *n* (percentage); * Median (25th–75th).

**Table 3 nutrients-13-02705-t003:** Relative abundance of specific bacterial groups in stools of BC patients and control group by specific primers per bacteria group /species.

Bacterial Population	Healthy Controls (*n* = 29)	BC Patients (*n* = 25)	*p*
Bacteroidetes phylum	19.14 (16.27–22.75)	14.24 (12.52–19.21)	<0.001
Firmicutes phylum	15.46 (10.75–33.57]	36.10 (24.91–43.46)	0.003
*Lactobacillales* sp.	0.006 (0.004–0.015)	0.006 (0.004–0.02)	0.91
*Clostridium* cluster IV	6.1 (3.63–11.91)	12.02 (9.33–21.56)	0.003
*Faecalibacterium prausnitzii*	4.88 (2.93–7.48)	7.63 (4.25–10.77)	0.25
*Clostridium* cluster XIVa	5.97 (4.02–15.59)	18.64 (10.94–22.43)	0.012
*R. intestinalis*	1.16 (0.86–1.96)	1.21 (0.64–2.11)	0.73
*Blautia* sp.	1.19 (0.99–1.79)	1.90 (1.28–2.37)	0.045
*Lactonifactor longoviformis*	0.002 (0.001–0.003)	0.002 (0.001–0.004)	0.45
Actinobacteria phylum			
*Bifidobacterium* sp.	0.126 (0.061–0.293)	0.070 (0.016–0.162)	0.11
Coriobacteriaceae	0.328 (0.169–0.619)	0.23 (0.14–0.60)	0.65
*E. lenta*	0.011 (0.007–0.020)	0.009 (0.005–0.017)	0.55
Proteobacteria phylum			
*Escherichia + Shigella*	0.005 (0.000–0.013)	0.006 (0.001–0.018)	0.50

The results are expressed as median % (25th–75th). *p* value ≤ 0.05, significant difference between the two groups, Wilcoxon test.

## Data Availability

The data presented in this study are available on request to the corresponding author. The data are not publicly available due to ethical reasons.

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
