# Peer review of "Faecal Microbiota Composition Varies between Patients with Breast Cancer and Healthy Women: A Comparative Case-Control Study"

_nutrients, 2021, doi:10.3390/nu13082705_

Round 1
Reviewer 1 Report
The authors have addressed all my comments and concerns well.
Author Response
We would like to thank you very much for reviewing this work.
Reviewer 2 Report
The authors Bobin-Dubigeon et al in their research article titled "Fecal Microbiota composition varies between patients with breast cancer and healthy women: A comparative case control study" have investigated the fecal populations of bacteria among breast cancer patients and samples for the normal population. The authors have indicated the presence of a microbiota difference between cancer and normal patients.
There are however a few concerns that would need to be addressed by the authors:
- The age differences between the two cohorts poses an issue that would need to be addressed in greater details given that there are several studies showing how the microbiome patterns changes with age in mouse models and also in humans. Is it possible to look at data available from normal cohorts from other published studies and compare the analysis so as to get a well informed idea of the microbiome populations that are disease specific and not related to an ageing gut. It the same holds true in that case this would be precedent for using younger controls in other studies. It is quite difficult at this point to rule out the role of ageing from this analysis.
- The authors need to analyze with respect to specific breast cancer groups within the study. This will help annotate if the microbiome change is TNM classification dependent.
- Additionally, how this correlate to tissue from breast cancer patients where there are multiple studies in literature. Is it very different, excluding controls here would be interesting to investigate the role of only cancer in general of wrt to different classifying parameters. This would help establish the soundness of fecal and tissue microbiomes in breast cancer.
Author Response
We would like to thank you you very much for reviewing our work.
Please find enclosed a file containing our responses.
Yours sincerely

Reviewer 3 Report
The paper addresses a topic worthy of investigation; however, statistic should be improbed and discussion revised accordingly.
Some suggestions:
a) use a final mutlivariate analysis to point out the global differnce between the two groups.
b) perform a multiple regression or a correspondence analysis for each group, to point put the correlation between some input parameters and gut microbiota
c) why t-test? do data follow the requisites for parametric statistic?
Author Response
We would like to thank your for reviewing our work.
Please find enclosed a file containing our reponse.
Sincerely ypurs

Round 2
Reviewer 2 Report
The authors have provided justification and more imporntantly added the concerns to the manuscript regarding the concerns raised by the reviewer which is commended.
Reviewer 3 Report
The authors addressed my issues, thus the paper can be accepted for publication
This manuscript is a resubmission of an earlier submission. The following is a list of the peer review reports and author responses from that submission.
Round 1
Reviewer 1 Report
It is my pleasure to review this manuscript. This study aimed to characterize the faecal microbiota from early 19 stage breast cancer (BC) patients and healthy controls and found that Shannon index was statistically lower in the BC group, compared to controls and a reduction of microbial diversity, a relative enrichment in Firmicutes, Clostridium cluster XIVa, and Clostridium cluster IV, as well as a depletion in Bacteroidetes, Bifidobacterium sp., Odoribacter sp., Butyricimonas.sp., and Coprococcus sp., in patients diagnosed with early BC compared to healthy women. This work and findings are very essential for breast cancer prevention and control.
Overall, it is well written manuscripts and I have several minor comments and suggestions in the following:
- I am a little concerned on the power of this study with 30 healthy and 25 breast cancer patients. Could the authors provide the power for the main analysis?
- In the microbiome analysis, since age is a potential confounders, I would recommend adjusting age in the analysis for alpha diversity and taxa relative abundance analysis.
- A subgroup or stratification analysis on type of cancer might also be considered for microbiome analysis, any difference in different types of cancer?
- This study showed both abundance (3.1.1) and relative abundance (3.1.2), I would suggest showing only relative abundance, which is more meaningful. Meanwhile, I would suggest building a multivariable model (adjusting confounders) for each of these significant taxa. In this way, the results will be more convincible. A volcano plot will also be nice fit for the taxa relative abundance analysis.
- Beta diversity was not shown in this study.
- Figure 4 content was missing in the manuscript.
- A study population flow chart was missing.
- Line 379 to 382, “Authors should …”, is this from a review comments and you just copied and pasted in the current manuscript? Need to be very cautious on “copy” and “paste”, it might easily get your paper rejected by editor.
Reviewer 2 Report
Journal: Nutrients
Manuscript ID: nutrients-1177792
Title: Faecal microbiota composition varies between patients with breast cancer and healthy women: a comparative case-control study
Overview:
This is a well written paper evaluating the fecal microbiota composition in women with early stage, hormone receptor positive breast cancer vs healthy case controls. The authors conclude differences in fecal microbiota between breast cancer patients and health controls, however a number of possible confounders and methodological flaws have been identified.
Major Comments:
- Almost ¼ more patients (22%) are post-menopausal in your control group. Thus, this is not case controlled for age, arguably a very important balancing factor for menopausal status and expression of estrogen responsive genes. This accounts for less estrogen in the control group, and alteration of estrogen responsive element genes, and thus modification of gut microbiota. This is a design flaw. It also should be stated in the statistical analysis/design as to what factors were “case controlled”, as arguably, age is one of the most important factors in this specific cohort.
- OCP and menopausal hormonal therapy may be confounders in your control group. In this case, your BC patients have a diagnosis of ER+/HER2- breast cancer and have likely been discontinued off any HRT or estrogen therapy prior to their clinic visit.
- Clinical stage 0 patients are DCIS/LCIS patients. These patients are not classified as early stage “breast cancer” patients, as these are pre-cancerous lesions. Thus, comparing this to invasive lesions is circumspect. I would refrain from using the word “cancer” as a blanket statement to include DCIS/LCIS patients.
- No power calculation, or study size estimate is mentioned in the statistical methods. This is also not specifically noted as a limitation in the discussion. This is a small, 25 patient study. It is unclear how the authors arrived at this number. Low sample size likely affects balancing of cohorts and perceived differences in gut microbiota.
Minor Comments:
- Line 54 – “…due to antibiotherapy…” Did the authors mean antibiotic therapy pertaining to reference 9?
- Table 1 – please clarify what * or ** means as there is no footer.
- Line 291-292 – This states that “menopausal status was the same in the two groups”. This is not true. It is numerically, and proportionally different. It was statistically “the same” 9 out of every 10 times, and not the same 1 out of every 10 times. Given the small numbers in this study, it is hard to conclude that these are the same.
- Table 3
- For the species identified as “significant” on Wilcoxon analysis, there is considerable overlap of the median 25th and 75th quartile values, suggesting no difference between bacterial species. It is difficult to conclude the a .21 increase in bacteriodies or .4 increase in Firmicutes is correlated with breast cancer risk, regardless of the Wilcoxon analysis.